# Learning Intrinsic Sparse Structures within Long Short-Term Memory

**Wei Wen**[*], **Yiran Chen & Hai Li**
Electrical and Computer Engineering, Duke University
{wei.wen,yiran.chen,hai.li}@duke.edu

**Yuxiong He**[†], **Samyam Rajbhandari**[†], **Minjia Zhang**[†], **Wenhan Wang**[†], **Fang Liu**[§] **& Bin Hu**[§]
Business AI[†] and Bing[§], Microsoft
{yuxhe,samyamr,minjiaz,wenhanw,fangliu,binhu}@microsoft.com

## Abstract

Model compression is significant for the wide adoption of *Recurrent Neural Networks* (RNNs) in both user devices possessing limited resources and business clusters requiring quick responses to large-scale service requests. This work aims to learn structurally-sparse *Long Short-Term Memory* (LSTM) by reducing the sizes of *basic structures* within LSTM units, including *input updates, gates, hidden states, cell states and outputs*. Independently reducing the sizes of basic structures can result in inconsistent dimensions among them, and consequently, end up with invalid LSTM units. To overcome the problem, we propose *Intrinsic Sparse Structures* (ISS) in LSTMs. Removing a component of ISS will simultaneously decrease the sizes of all basic structures by one and thereby always maintain the dimension consistency. By learning ISS within LSTM units, the obtained LSTMs remain regular while having much smaller basic structures. Based on group Lasso regularization, our method achieves $10.59\times$ speedup without losing any perplexity of a language modeling of Penn TreeBank dataset. It is also successfully evaluated through a compact model with only 2.69M weights for machine Question Answering of SQuAD dataset. Our approach is successfully extended to non-LSTM RNNs, like Recurrent Highway Networks (RHNs). Our source code is available[1].

## 1 Introduction

*Model Compression* (Jaderberg et al. (2014), Han et al. (2015a), Wen et al. (2017), Louizos et al. (2017)) is a class of approaches of reducing the size of *Deep Neural Networks* (DNNs) to accelerate inference. *Structure Learning* (Zoph & Le (2017), Philipp & Carbonell (2017), Cortes et al. (2017)) emerges as an active research area for DNN structure exploration, potentially replacing human labor with machine automation for design space exploration. In the intersection of both techniques, an important area is to learn compact structures in DNNs for efficient inference computation using minimal memory and execution time without losing accuracy. Learning compact structures in *Convolutional Neural Networks* (CNNs) have been widely explored in the past few years. Han et al. (2015b) proposed connection pruning for sparse CNNs. Pruning method also works successfully in coarse-grain levels, such as pruning filters in CNNs (Li et al. (2017)) and reducing neuron numbers (Alvarez & Salzmann (2016)). Wen et al. (2016) presented a general framework to learn versatile compact structures (neurons, filters, filter shapes, channels and even layers) in DNNs.

Learning the compact structures in *Recurrent Neural Networks* (RNNs) is more challenging. As a recurrent unit is shared across all the time steps in sequence, compressing the unit will aggressively affect all the steps. A recent work by Narang et al. (2017) proposes a pruning approach that deletes up to $90\%$ connections in RNNs. Connection pruning methods sparsify weights of recurrent units but cannot explicitly change *basic structures*, *e.g.*, the number of *input updates, gates,*

---

[*]Major work was done as an intern in Microsoft Research and Bing.

[1]https://github.com/wenwei202/iss-rnns

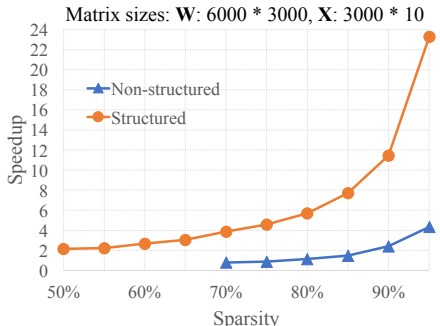 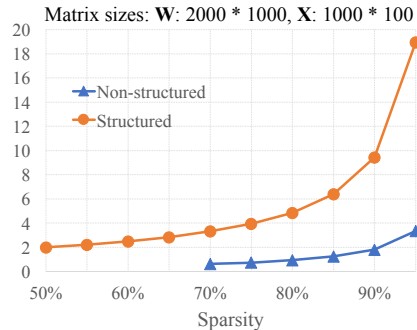

Figure 1: Speedups of matrix multiplication using non-structured and structured sparsity. Speeds are measured in Intel MKL implementations in Intel Xeon CPU E5-2673 v3 @ 2.40GHz. General matrix-matrix multiplication (GEMM) of $\mathbf{W} \cdot \mathbf{X}$ is implemented by `cblas_sgemm`. The matrix sizes are selected to reflect commonly used GEMMs in LSTMs. For example, (a) represents GEMM in LSTMs with hidden size 1500, input size 1500 and batch size 10. To accelerate GEMM by sparsity, $\mathbf{W}$ is sparsified. In non-structured sparsity approach, $\mathbf{W}$ is randomly sparsified and encoded as Compressed Sparse Row format for sparse computation (using `mkl_scsrmm`); in structured sparsity approach, $2k$ columns and $4k$ rows in $\mathbf{W}$ are removed to match the same level of sparsity (*i.e.,* the percentage of removed parameters) for faster GEMM under smaller sizes.

*hidden states, cell states and outputs*. Moreover, the obtained sparse matrices have an irregular/non-structured pattern of non-zero weights, which is unfriendly for efficient computation in modern hardware systems (Lebedev & Lempitsky (2016)). Previous study (Wen et al. (2016)) on sparse matrix multiplication in GPUs showed that the speedup[2] was either counterproductive or ignorable. More specific, with sparsity[3] of 67.6%, 92.4%, 97.2%, 96.6% and 94.3% in weight matrices of *AlexNet*, the speedup was $0.25\times$, $0.52\times$, $1.38\times$, $1.04\times$, and $1.36\times$, respectively. This problem also exists in CPUs. Fig. 1 shows that non-structured pattern in sparsity limits the speedup. We only starts to observe speed gain when the sparsity is beyond 80%, and the speedup is about $3\times$ to $4\times$ even when the sparsity is 95% which is far below the theoretical $20\times$. In this work, we focus on learning structurally sparse LSTMs for computation efficiency. More specific, we aim to reduce the number of basic structures simultaneously during learning, such that the obtained LSTMs have the original schematic with dense connections but with smaller sizes of these basic structures. Such compact models have structured sparsity, with columns and rows in weight matrices removed, whose computation efficiency is shown in Fig. 1. Moreover, off-the-shelf libraries in deep learning frameworks can be directly utilized to deploy the reduced LSTMs. Details should be explained.

There is a vital challenge originated from recurrent units: as the basic structures interweave with each other, independently removing these structures can result in mismatch of their dimensions and then inducing invalid recurrent units. The problem does not exist in CNNs, where neurons (or filters) can be independently removed without violating the usability of the final network structure. One of our key contributions is to identify the structure inside RNNs that shall be considered as a group to most effectively explore sparsity in basic structures. More specific, we propose *Intrinsic Sparse Structures* (ISS) as groups to achieve the goal. By removing weights associated with one component of ISS, the sizes/dimensions (of basic structures) are *simultaneously* reduced by one.

We evaluated our method by LSTMs and RHNs in language modeling of Penn Treebank dataset (Marcus et al. (1993)) and machine Question Answering of SQuAD dataset (Rajpurkar et al. (2016)). Our approach works both in fine-tuning and in training from scratch. In a RNN with two stacked LSTM layers with hidden sizes of 1500 (*i.e.,* 1500 components of ISS) for language modeling (Zaremba et al. (2014)), our method learns that the sizes of 373 and 315 in the first and second LSTMs, respectively, are sufficient for the same perplexity. It achieves $10.59\times$ speedup of inference time. The result is obtained by training from scratch with the same number of epochs. Directly training LSTMs with sizes of 373 and 315 cannot achieve the same perplexity, which proves the advantage of learning ISS for model compression. Encouraging results are also obtained in more

---

[2]Defined as (new speed)/(original speed).

[3]Defined as the percentage of zeros.

compact and state-of-the-art models – the RHN models (Zilly et al. (2017)) and BiDAF model (Seo et al. (2017)).

## 2  RELATED WORK

A major approach in DNN compression is to reduce the complexity of structures within DNNs. The studies can be categorized to three classes: removing redundant structures in original DNNs, approximating the original function of DNNs (Denil et al. (2013), Jaderberg et al. (2014), Hinton et al. (2015), Lu et al. (2016), Prabhavalkar et al. (2016), Molchanov et al. (2017)), and designing DNNs with inherently compact structures (Szegedy et al. (2015), He et al. (2016), Wu et al. (2017), Bradbury et al. (2016)). Our method belongs to the first category.

Research on removing redundant structures in *Feed-forward Neural Networks* (FNNs), typically in CNNs, has been extensively studied. Based on $\ell_1$ regularization (Liu et al. (2015), Park et al. (2017)) or connection pruning (Han et al. (2015b), Guo et al. (2016)), the number of connections/parameters can be dramatically reduced. Group Lasso based methods were proved to be effective in reducing coarse-grain structures (*e.g.,* neurons, filters, channels, filter shapes, and even layers) in CNNs (Wen et al. (2016), Alvarez & Salzmann (2016), Lebedev & Lempitsky (2016), Yoon & Hwang (2017)). For instance, Wen et al. (2016) reduced the number of layers from 32 to 18 in *ResNet* without any accuracy loss for CIFAR-10 dataset. A recent work by Narang et al. (2017) advances connection pruning techniques for RNNs. It compresses the size of *Deep Speech 2* (Amodei et al. (2016)) from 268 MB to around 32 MB. However, to the best of our knowledge, little work has been carried out to reduce coarse-grain structures beyond fine-grain connections in RNNs. To fill this gap, our work targets to develop a method that can learn to reduce the number of basic structures within LSTM units. After learning those structures, final LSTMs are still regular LSTMs with the same connectivity, but have the sizes reduced.

Another line of related research is *Structure Learning* of FNNs or CNNs. Zoph & Le (2017) uses reinforcement learning to search good neural architectures. Philipp & Carbonell (2017) dynamically adds and eliminates neurons in FNNs by using group Lasso regularization. Cortes et al. (2017) gradually adds sub-networks to current networks to incrementally reduce the objective function. All these works focused on finding optimal structures in FNNs or CNNs for classification accuracy. In contrast, this work aims at learning compact structures in LSTMs for model compression.

## 3  LEARNING INTRINSIC SPARSE STRUCTURES

### 3.1  INTRINSIC SPARSE STRUCTURES

The computation within LSTMs is (Hochreiter & Schmidhuber (1997))

$$
\begin{aligned}
\mathbf{i}_t &= \sigma\left(\mathbf{x}_t \cdot \mathbf{W}_{xi} + \mathbf{h}_{t-1} \cdot \mathbf{W}_{hi} + \mathbf{b}_i\right) \\
\mathbf{f}_t &= \sigma\left(\mathbf{x}_t \cdot \mathbf{W}_{xf} + \mathbf{h}_{t-1} \cdot \mathbf{W}_{hf} + \mathbf{b}_f\right) \\
\mathbf{o}_t &= \sigma\left(\mathbf{x}_t \cdot \mathbf{W}_{xo} + \mathbf{h}_{t-1} \cdot \mathbf{W}_{ho} + \mathbf{b}_o\right) \\
\mathbf{u}_t &= tanh\left(\mathbf{x}_t \cdot \mathbf{W}_{xu} + \mathbf{h}_{t-1} \cdot \mathbf{W}_{hu} + \mathbf{b}_u\right) \\
\mathbf{c}_t &= \mathbf{f}_t \odot \mathbf{c}_{t-1} + \mathbf{i}_t \odot \mathbf{u}_t \\
\mathbf{h}_t &= \mathbf{o}_t \odot tanh\left(\mathbf{c}_t\right)
\end{aligned}
\tag{1}
$$

where $\odot$ is element-wise multiplication, $\sigma(\cdot)$ is sigmoid function, and $tanh(\cdot)$ is hyperbolic tangent function. Vectors are row vectors. $\mathbf{W}$s are weight matrices, which transform the concatenation (of hidden states $\mathbf{h}_{t-1}$ and inputs $\mathbf{x}_t$) to input updates $\mathbf{u}_t$ and gates ($\mathbf{i}_t$, $\mathbf{f}_t$ and $\mathbf{o}_t$). Fig. 2 is the schematic of LSTMs in the layout of Olah (2015). The transformations by $\mathbf{W}$s and the corresponding nonlinear functions are illustrated in rectangle blocks. Our goal is to reduce the size of this sophisticated structure within LSTMs, meanwhile maintaining the original schematic. Because of element-wise operators ("$\oplus$" and "$\otimes$"), all vectors along the blue band in Fig. 2 must have the same dimension. We call this constraint as "*dimension consistency*". The vectors required to obey the dimension consistency include input updates, all gates, hidden states, cell states, and outputs. Note that hidden states are usually outputs connected to classifier layer or stacked LSTM layers. As can be seen in Fig. 2, vectors (along the blue band) interweave with each other so removing an individual component from one or a few vectors *independently* can result in the violation of dimension consistency.

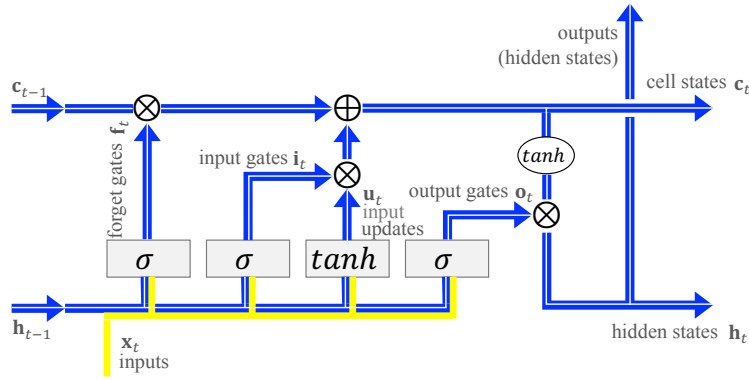

Figure 2: Intrinsic Sparse Structures (ISS) in LSTM units.

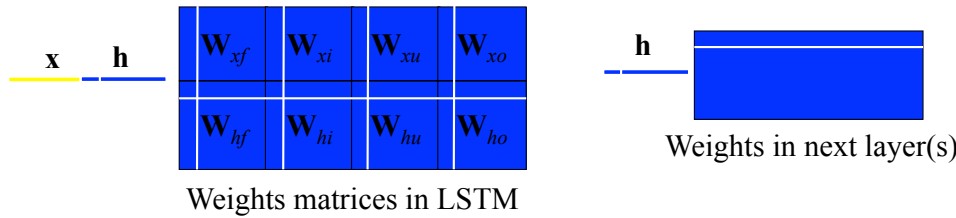

Figure 3: Applying Intrinsic Sparse Structures in weight matrices.

To overcome this, we propose *Intrinsic Sparse Structures* (ISS) within LSTMs as shown by the blue band in Fig. 2. One component of ISS is highlighted as the white strip. By decreasing the size of ISS (*i.e.*, the width of the blue band), we are able to simultaneously reduce the dimensions of basic structures.

To learn sparse ISS, we turn to weight sparsifying. There are totally eight weight matrices in Eq. (1). We organize them in the form of Fig. 3 as basic LSTM cells in TensorFlow. We can remove one component of ISS by zeroing out all associated weights in the white rows and white columns in Fig. 3. Why? Suppose the $k$-th hidden state of $\mathbf{h}$ is removable, then the $k$-th row in the lower four weight matrices can be all zeros (as shown by the left white horizontal line in Fig. 3), because those weights are on connections receiving the $k$-th *useless* hidden state. Likewise, all connections receiving the $k$-th hidden state in next layer(s) can be removed as shown by the right white horizontal line. Note that next layer(s) can be an output layer, LSTM layers, fully-connected layers, or a mix of them. ISS overlay two or more layers, without explicit explanation, we refer to the first LSTM layer as the ownership of ISS. When the $k$-th hidden state turns useless, the $k$-th output gate and $k$-th cell state generating this hidden state are removable. As the $k$-th output gate is generated by the $k$-th column in $\mathbf{W}_{xo}$ and $\mathbf{W}_{ho}$, these weights can be zeroed out (as shown by the fourth vertical white line in Fig. 3). Tracing back against the computation flow in Fig. 2, we can reach similar conclusions for forget gates, input gates and input updates, as respectively shown by the first, second and third vertical line in Fig. 3. For convenience, we call the weights in white rows and columns as an "*ISS weight group*". Although we propose ISS in LSTMs, variants of ISS for vanilla RNNs, *Gated Recurrent Unit* (GRU) (Cho et al. (2014)), and *Recurrent Highway Networks* (RHNs) (Zilly et al. (2017)) can also be realized based on the same philosophy.

For even a medium-scale LSTM, the number of weights in one ISS weight group can be very large. It seems to be very aggressive to simultaneously slaughter so many weights to maintain the original recognition performance. However, the proposed ISS intrinsically exists within LSTMs and can even be unveiled by *independently* sparsifying each weight using $\ell_1$-norm regularization. The experimental result is covered in Appendix A. It unveils that sparse ISS *intrinsically* exist in LSTMs and the learning process can easily converge to the status with a high ratio of ISS removed. In Section 3.2, we propose a learning method to *explicitly* remove much more ISS than the implicit $\ell_1$-norm regularization.

## 3.2 LEARNING METHOD

Suppose $\mathbf{w}_k^{(n)}$ is a vector of all weights in the $k$-th component of ISS in the $n$-th LSTM layer ($1 \leq n \leq N$ and $1 \leq k \leq K^{(n)}$), where $N$ is the number of LSTM layers and $K^{(n)}$ is the number of ISS components (*i.e.*, hidden size) of the $n$-th LSTM layer. The optimization goal is to remove as many "ISS weight groups" $\mathbf{w}_k^{(n)}$ as possible without losing accuracy. Methods to remove weight groups (such as filters, channels and layers) have been successfully studied in CNNs as summarized in Section 2. However, how these methods perform in RNNs is unknown. Here, we extend the group Lasso based methods (Yuan & Lin (2006)) to RNNs for ISS sparsity learning. More specific, the group Lasso regularization is added to the minimization function in order to encourage sparsity in ISS. Formally, the ISS regularization is

$$R(\mathbf{w}) = \sum_{n=1}^{N} \sum_{k=1}^{K^{(n)}} \left|\left|\mathbf{w}_k^{(n)}\right|\right|_2,$$ (2)

where $\mathbf{w}$ is the vector of all weights and $|| \cdot ||_2$ is $\ell_2$-norm (*i.e.*, Euclidean length). In *Stochastic Gradient Descent* (SGD) training, the step to update each ISS weight group becomes

$$\mathbf{w}_k^{(n)} \leftarrow \mathbf{w}_k^{(n)} - \eta \cdot \left( \frac{\partial E(\mathbf{w})}{\partial \mathbf{w}_k^{(n)}} + \lambda \cdot \frac{\mathbf{w}_k^{(n)}}{\left|\left|\mathbf{w}_k^{(n)}\right|\right|_2} \right),$$ (3)

where $E(\mathbf{w})$ is data loss, $\eta$ is learning rate and $\lambda > 0$ is the coefficient of group Lasso regularization to trade off recognition accuracy and ISS sparsity. The regularization gradient, *i.e.*, the last term in Eq. (3), is a unit vector. It constantly squeezes the Euclidean length of each $\mathbf{w}_k^{(n)}$ to zero, such that, a high portion of ISS components can be enforced to fully-zeros after learning. To avoid division by zero in the computation of regularization gradient, we can add a tiny number $\epsilon$ in $|| \cdot ||_2$, that is,

$$\left|\left|\mathbf{w}_k^{(n)}\right|\right|_2 \triangleq \sqrt{\epsilon + \sum_j \left( w_{kj}^{(n)} \right)^2},$$ (4)

where $w_{kj}^{(n)}$ is the $j$-th element of $\mathbf{w}_k^{(n)}$. We set $\epsilon = 1.0e - 8$. The learning method can effectively squeeze many groups near zeros, but it is very hard to exactly stabilize them as zeros because of the always-present fluctuating weight updates. Fortunately, the fluctuation is within a tiny ball centered at zero. To stabilize the sparsity during training, we zero out the weights whose absolute values are smaller than a pre-defined threshold $\tau$. The process of thresholding is applied per mini-batch.

## 4 EXPERIMENTS

Our experiments use published models as baselines. The application domains include language modeling of Penn TreeBank and machine Question Answering of SQuAD dataset. For more comprehensive evaluation, we sparsify ISS in LSTM models with both a large hidden size of $1500$ and a small hidden size of $100$. We also extended ISS approach to state-of-the-art *Recurrent Highway Networks* (RHNs) (Zilly et al. (2017)) to reduce the number of units per layer. We maximize threshold $\tau$ to fully exploit the benefit. For a specific application, we preset $\tau$ by cross validation. The maximum $\tau$ which sparsifies the dense model (baseline) without deteriorating its performance is selected. The validation of $\tau$ is performed only once and no training effort is needed. $\tau$ is $1.0e - 4$ for the stacked LSTMs in Penn TreeBank, and it is $4.0e - 4$ for the RHN and the BiDAF model. We used HyperDrive by Rasley et al. (2017) to explore the hyperparameter of $\lambda$. More details can be found in our source code.

To measure the inference speed, the experiments were run on a dual socket Intel Xeon CPU E5-2673 v3 @ 2.40GHz processor with a total of 24 cores (12 per socket) and 128GB of memory. Intel MKL library 2017 update 2 was used for matrix-multiplication operations. OpenMP runtime was utilized for parallelism. We used Intel C++ Compiler 17.0 to generate executables that were run on Windows Server 2016. Each of the experiments was run for 1000 iterations, and the execution time was averaged to find the execution latency.

Table 1: Learning ISS sparsity from scratch in stacked LSTMs.

| Method | Dropout keep ratio | Perplexity (validate, test) | ISS # in (1st , 2nd) LSTM | Weight # | Total time* | Speedup | Mult-add reduction[†] |
|---|---|---|---|---|---|---|---|
| baseline | 0.35 | (82.57, 78.57) | (1500, 1500) | 66.0M | 157.0ms | 1.00× | 1.00× |
| ISS | 0.60 | (82.59, 78.65) | (373, 315) | 21.8M | 14.82ms | 10.59× | 7.48× |
| | | (80.24, 76.03) | (381, 535) | 25.2M | 22.11ms | 7.10× | 5.01× |
| direct design | 0.55 | (90.31, 85.66) | (373, 315) | 21.8M | 14.82ms | 10.59× | 7.48× |

* Measured with 10 batch size and 30 unrolled steps.
[†] The reduction of multiplication-add operations in matrix multiplication. Defined as (original Mult-add)/(left Mult-add)

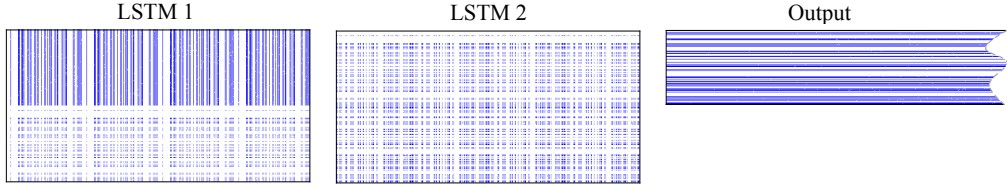

Figure 4: Intrinsic Sparse Structures learned by group Lasso regularization (zoom in for better view). Original weight matrices are plotted, where blue dots are nonzero weights and white ones refer zeros. For better visualization, original matrices are evenly down-sampled by $10 \times 10$.

## 4.1 LANGUAGE MODELING

### 4.1.1 STACKED LSTMS

A RNN with two stacked LSTM layers for language modeling (Zaremba et al. (2014)) is selected as the baseline. It has hidden sizes of 1500 (*i.e.*, 1500 components of ISS) in both LSTM units. The output layer has a vocabulary of 10000 words. The dimension of word embedding in the input layer is 1500. Word embedding layer is not sparsified because the computation of selecting a vector from a matrix is very efficient. The same training scheme as the baseline is adopted to learn ISS sparsity, except a larger dropout keep ratio of 0.6 versus 0.35 of the baseline because group Lasso regularization can also avoid over-fitting. All models are trained from scratch for 55 epochs. The results are shown in Table 1. Note that, when trained using dropout keep ratio of 0.6 without adopting group Lasso regularization, the baseline over-fits and the lowest validation perplexity is 97.73. The trade-off of perplexity and sparsity is controlled by $\lambda$. In the second row, with tiny perplexity difference from baseline, our approach can reduce the number of ISS in the first and second LSTM unit from 1500, down to 373 and 315, respectively. It reduces the model size from 66.0M to 21.8M and achieves 10.59× speedup. Remarkably, the practical speedup (10.59×) even goes beyond theoretical mult-add reduction (7.48×) as shown in Table 1 —which comes from the increased computational efficiency. When applying structured sparsity, the underlying weight matrices become smaller so as to fit into the L3 cache with good locality, which improves the FLOPS (floating point operations per second). This is a key advantage of our approach over non-structurally sparse RNNs generated by connection pruning (Narang et al. (2017)), which suffers from irregular memory access pattern and inferior-theoretical speedup. At last, when learning a compact structure, our method can perform as structure regularization to avoid overfitting. As shown in the third row in Table 1, lower perplexity is achieved by even a smaller (25.2M) and faster (7.10×) model. Its learned weight matrices are visualized in Fig. 4, where 1119 and 965 ISS components shown by white strips are removed in the first and second LSTM, respectively.

A straightforward way to reduce model complexity is to directly design a RNN with a smaller hidden size and train from scratch. Compare with direct design approach, our ISS method can automatically learn optimal structures within LSTMs. More importantly, compact models learned by ISS method have lower perplexity, comparing with direct design method. To evaluate it, we directly design a RNN with exactly the same structure of the second RNN in Table 1 and train it from scratch instead of learning ISS from a larger RNN. The result is included in the last row of Table 1. We tuned dropout keep ratio to get best perplexity for the directly-designed RNN. The final test perplexity is 85.66, which is 7.01 higher that our ISS method.

Table 2: Learning ISS sparsity from scratch in RHNs.

| Method | $\lambda$ | Perplexity (validate, test) | RHN width | Parameter # |
|--------|------|------|------|------|
| baseline | 0.0 | (67.9, 65.4) | 830 | 23.5M |
| ISS[*] | 0.004 | (67.5, 65.0) | 726 | 18.9M |
| ISS[*] | 0.005 | **(68.1, 65.4)** | **517** | **11.1M** |
| ISS[*] | 0.006 | (70.3, 67.7) | 403 | 7.6M |
| ISS[*] | 0.007 | (74.5, 71.2) | 328 | 5.7M |

[*] All dropout ratios are multiplied by $0.6\times$.

### 4.1.2 EXTENSION TO RECURRENT HIGHWAY NETWORKS

*Recurrent Highway Networks* (RHN) (Zilly et al. (2017)) is a class of state-of-the-art recurrent models, which enable "step-to-step transition depths larger than one". In a RHN, we define the number of units per layer as *RHN width*. Specifically, we select the "Variational RHN + WT" model in Table 1 of Zilly et al. (2017) as the baseline. It has depth 10 and width 830, with totally 23.5M parameters. In a nutshell, our approach can reduce the RHN width from 830 to 517 without losing perplexity.

Following the same idea of identifying the "ISS weight groups" to reduce the size of basic structures in LSTMs, we can identify the groups in RHNs to reduce the RHN width. In brief, one group include corresponding columns/rows in weight matrices of the $H$ nonlinear transform, of the $T$ and $C$ gates, and of the embedding and output layers. The group size is 46520. The groups are indicated by JSON files in our source code[4]. By learning ISS in RHNs, we can simultaneously reduce the dimension of word embedding and the number of units per layer.

Table 2 summarizes results. All experiments are trained from scratch with the same hyperparameters in the baseline, except that smaller dropout ratios are used in ISS learning. Larger $\lambda$, smaller RHN width but higher perplexity. More importantly, without losing perplexity, our approach can learn a smaller model with RHN width 517 from an initial model with RHN width 830. This reduces the model size to 11.1M, which is $52.8\%$ reduction. Moreover, ISS learning can find a smaller RHN model with width 726, meanwhile improve the state-of-the-art perplexity as shown by the second entry in Table 2.

### 4.2 MACHINE READING COMPREHENSION

We evaluate ISS method by state-of-the-art dataset (SQuAD) and model (BiDAF). SQuAD (Rajpurkar et al. (2016)) is a recently released reading comprehension dataset, crowdsourced from $100,000+$ question-answer pairs on $500+$ Wikipedia articles. ExactMatch (EM) and F1 scores are two major metrics for the task[5]. The higher those scores are, the better the model is. We adopt BiDAF (Seo et al. (2017)) to evaluate how ISS method works in small LSTM units. BiDAF is a compact machine Question Answering model with totally 2.69M weights. The ISS sizes are only 100 in all LSTM units. The implementation of BiDAF is made available by its authors [6].

BiDAF has character, word and contextual embedding layers to extract representations from input sentences, following which are bi-directional attention layer, modeling layer, and final output layer. LSTM units are used in contextual embedding layer, modeling layer, and output layer. All LSTMs are bidirectional (Schuster & Paliwal (1997)). In a bidirectional LSTM, there are one forward plus one backward LSTM branch. The two branches share inputs and their outputs are concatenated for next stacked layers. We found that it is hard to remove ISS components in contextual embedding layer, because the representations are relatively dense as it is close to inputs and the original hidden size (100) is relatively small. In our experiments, we exclude LSTMs in contextual embedding layer and sparsify all other LSTM layers. Those LSTM layers are the computation bottleneck of BiDAF.

---

[4] groups_hidden830.json

[5] Refer to Rajpurkar et al. (2016) for the definition of EM and F1 scores.

[6] https://github.com/allenai/bi-att-flow/tree/dev

Table 3: Remaining ISS components in BiDAF by fine-tuning.

| EM | F1 | ModFwd1 | ModBwd1 | ModFwd2 | ModBwd2 | OutFwd | OutBwd | weight # | Total time[*] |
|---|---|---|---|---|---|---|---|---|---|
| 67.98 | 77.85 | 100 | 100 | 100 | 100 | 100 | 100 | 2.69M | 6.20ms |
| 67.21 | 76.71 | 100 | 95 | 78 | 82 | 71 | 52 | 2.08M | 5.79ms |
| 66.59 | 76.40 | 84 | 90 | 38 | 46 | 34 | 21 | 1.48M | 4.52ms |
| 65.29 | 75.47 | 54 | 47 | 22 | 30 | 18 | 12 | 1.03M | 3.54ms |
| 64.81 | 75.22 | 52 | 50 | 19 | 26 | 15 | 12 | 1.01M | 3.51ms |

[*] Measured with batch size 1.

Table 4: Remaining ISS components in BiDAF by training from scratch.

| EM | F1 | ModFwd1 | ModBwd1 | ModFwd2 | ModBwd2 | OutFwd | OutBwd | weight # | Total time[*] |
|---|---|---|---|---|---|---|---|---|---|
| 67.98 | 77.85 | 100 | 100 | 100 | 100 | 100 | 100 | 2.69M | 6.20ms |
| 67.36 | 77.16 | 87 | 81 | 87 | 92 | 74 | 96 | 2.29M | 5.83ms |
| 66.32 | 76.22 | 51 | 33 | 42 | 58 | 37 | 26 | 1.17M | 4.46ms |
| 65.36 | 75.78 | 20 | 33 | 40 | 38 | 31 | 16 | 0.95M | 3.59ms |
| 64.60 | 74.99 | 23 | 22 | 35 | 35 | 25 | 14 | 0.88M | 2.74ms |

[*] Measured with batch size 1.

We profiled the computation time on CPUs, and find those LSTM layers (excluding contextual embedding layer) consume $76.47\%$ of total inference time. There are three bi-directional LSTM layers we will sparsify, two of which belong to the modeling layer, and one belongs to the output layer. More details of BiDAF are covered by Seo et al. (2017). For brevity, we mark the forward (backward) path of the 1st bi-directional LSTM in the modeling layer as `ModFwd1` (`ModBwd1`). Similarly, `ModFwd2` and `ModBwd2` are for the 2nd bi-directional LSTM. Forward (backward) LSTM path in the output layer are marked as `OutFwd` and `OutBwd`.

As discussed in Section 3.1, multiple parallel layers can receive the hidden states from the same LSTM layer and all connections (weights) receive those hidden states belong to the same ISS. For instance, `ModFwd2` and `ModBwd2` both receive hidden states of `ModFwd1` as inputs, therefore the $k$-th "ISS weight group" includes the $k$-th rows of weights in both `ModFwd2` and `ModBwd2`, plus the weights in the $k$-th ISS component within `ModFwd1`. For simplicity, we use "ISS of `ModFwd1`" to refer to the whole group of weights. Structures of six ISS are included in Table 5 in Appendix B. We learn ISS sparsity in BiDAF by both fine-tuning the baseline and training from scratch. All the training schemes keep as the same as the baseline except applying a higher dropout keep ratio. After training, we zero out weights whose absolute values are smaller than $0.02$. This does not impact EM and F1 scores, but increase sparsity.

Table 3 shows the EM, F1, the number of remaining ISS components, model size, and inference speed. The first row is the baseline BiDAF. Other rows are obtained by fine-tuning baseline using ISS regularization. In the second row by learning ISS, with small EM and F1 loss, we can reduce ISS in all LSTMs except `ModFwd1`. For example, almost half of the ISS components are removed in `OutBwd`. By increasing the strength of group Lasso regularization ($\lambda$), we can increase the ISS sparsity by losing some EM/F1 scores. The trade-off is listed in Table 3. With 2.63 F1 score loss, the sizes of `OutFwd` and `OutBwd` can be reduced from original 100 to 15 and 12, respectively. At last, we find it hard to reduce ISS sizes without losing any EM/F1 score. This implies that BiDAF is compact enough and its scale is suitable for both computation and accuracy. However, our method can still significantly compress this compact model under acceptable performance loss.

At last, instead of fine-tuning baseline, we train BiDAF from scratch with ISS learning. The results are summarized in Table 4. Our approach also works well when training from scratch. Overall, training from scratch balances the sparsity across all layers better than fine-tuning, which results in even better compression of model size and speedup of inference time. The histogram of vector lengths of "ISS weight groups" is plotted in Appendix C.

## 5 CONCLUSION

We proposed *Intrinsic Sparse Structures* (ISS) within LSTMs and its learning method to simultaneously reduce the sizes of input updates, gates, hidden states, cell states and outputs within the

sophisticated LSTM structure. By learning ISS, a structurally sparse LSTM can be obtained, which essentially is a regular LSTM with reduced hidden dimension. Thus, no software or hardware specific customization is required to get storage saving and computation acceleration. Though ISS is proposed with LSTMs, it can be easily extended to vanilla RNNs, *Gated Recurrent Unit* (GRU) (Cho et al. (2014)), and *Recurrent Highway Networks* (RHNs) (Zilly et al. (2017)).

ACKNOWLEDGMENTS

Thank researchers and engineers in Microsoft for giving valuable feedback on this work, with acknowledgments to Wei He, Freddie Zhang, Yi Liu, Jacob Devlin and Chen Zhou. Also thank Jeff Rasley (intern in Microsoft Research, Brown University) for helping me to use HyperDrive (Rasley et al. (2017)) for hyper-parameter exploration. This work was supported in part by NSF CCF-1744082, NSF CCF-1725456 and DOE SC0017030. Any opinions, findings, conclusions or recommendations expressed in this material are those of the authors and do not necessarily reflect the views of NSF, DOE, or their contractors.

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

## APPENDIX A   ISS UNVEILED BY $\ell_1$-NORM REGULARIZATION

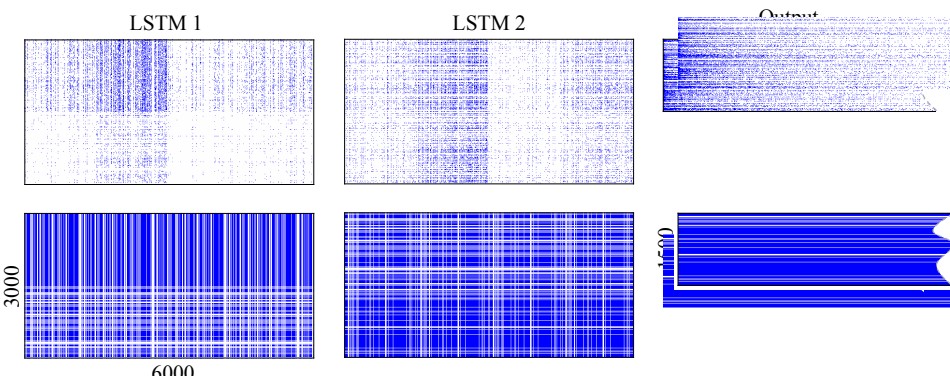

Figure 5: Intrinsic Sparse Structures unveiled by $\ell_1$ regularization (zoom in for a better view). The top row shows the original weight matrices, where blue dots are nonzero weights and white ones refer zeros; the bottom row are the weight matrices in the format of Fig. 3, where white strips are ISS components whose weights are all zeros. For better visualization, the original matrices are evenly down-sampled by $10 \times 10$.

We take the large stacked LSTMs by Zaremba et al. (2014) for language modeling as the example. The network has two stacked LSTM layers whose dimensions of inputs and states are both 1500, and it has an output layer with a vocabulary of 10000 words. The sizes of "ISS weight groups" of two LSTM layers are 24000 and 28000. The perplexities of validation set and test set are respectively 82.57 and 78.57. We fine-tune this baseline LSTMs with $\ell_1$-norm regularization. The same training hyper-parameters as the baseline are adopted, except a bigger dropout keep ratio of 0.6 (original 0.35). A weaker dropout is used because $\ell_1$-norm is also a regularization to avoid overfitting. A too strong dropout plus $\ell_1$-norm regularization can result in underfitting. The weight decay of $\ell_1$-norm regularization is 0.0001. The sparsified network has validation perplexity and test perplexity of 82.40 and 78.60, respectively, which is approximately the same with the baseline. The sparsity of weights in the first LSTM layer, the second LSTM layer and the last output layer is 91.66%, 90.32% and 90.22%, respectively. Fig. 5 plots the learned sparse weight matrices. The sparse matrices in the top row reveal some interesting patterns: there are lots of all-zero columns and rows, and their positions are highly correlated. Those patterns are profiled in the bottom row. Much to our surprise, sparsifying individual weight *independently* can converge to sparse LSTMs with many ISS removed—504 and 220 ISS components in the first and second LSTM layer are all-zeros.

## APPENDIX B   ISS IN BIDAF

Table 5:  The ISS in BiDAF.

| LSTM name | Dimensions of weight matrix | Receivers of hidden states | Size of "ISS weight group" |
|---|---|---|---|
| `ModFwd1` | $900 \times 400$ | `ModFwd2` `ModBwd2` | 4800 |
| `ModBwd1` | $900 \times 400$ | `ModFwd2` `ModBwd2` | 4800 |
| `ModFwd2` | $300 \times 400$ | `OutFwd` `OutBwd` logit layer for start index | 3201 |
| `ModBwd2` | $300 \times 400$ | `OutFwd` `OutBwd` logit layer for start index | 3201 |
| `OutFwd` | $1500 \times 400$ | logit layer for end index | 6401 |
| `OutBwd` | $1500 \times 400$ | logit layer for end index | 6401 |

## APPENDIX C   HISTOGRAM OF VECTOR LENGTHS OF ISS WEIGHT GROUPS IN BIDAF

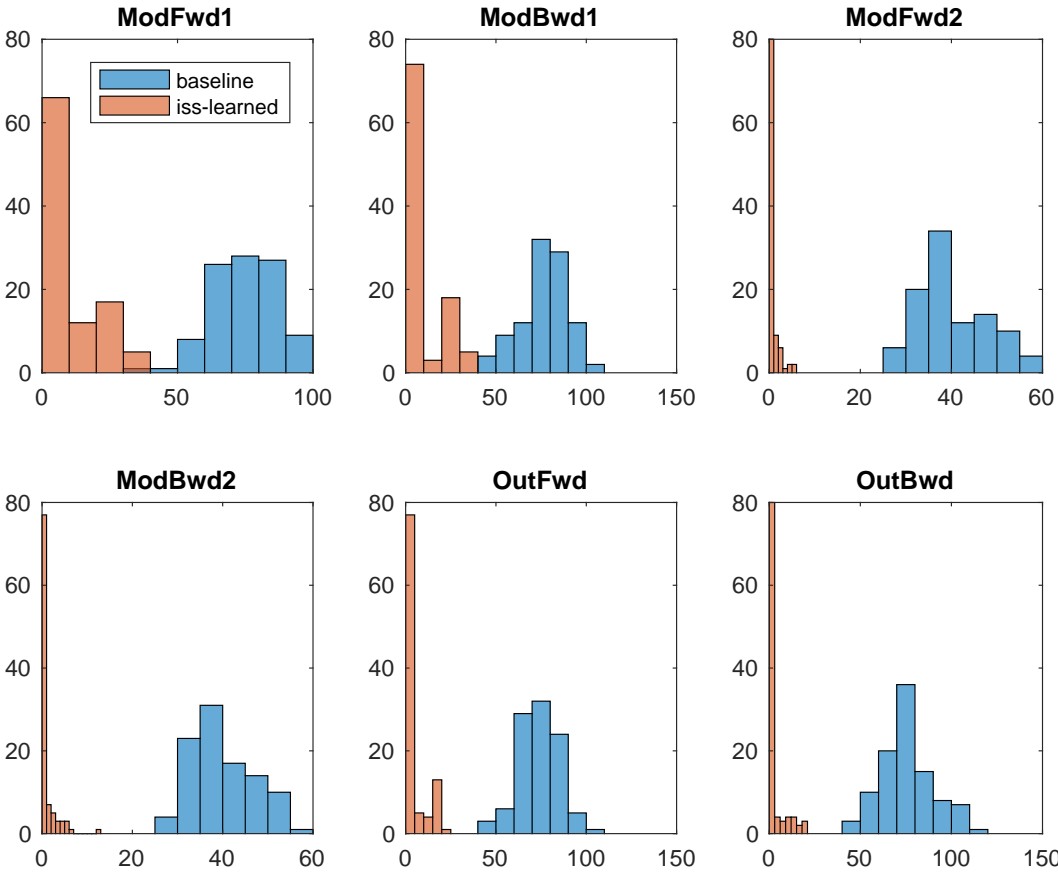

Figure 6: Histogram of vector lengths of "ISS weight groups" in BiDAF. The ISS-learned BiDAF is the one in the third row of Table 4 with EM 66.32 and F1 76.22. Using our approach, the lengths are regularized closer to zeros with a peak at the zero, resulting in high ISS sparsity.

