# OpenReview forum: "Learning Intrinsic Sparse Structures within Long Short-Term Memory"
_ICLR.cc/2018/Conference — Accept (Poster)_

### Official Review · AnonReviewer2 · 2017-11-21
**Nice work on RNN compression**

**Rating:** 7
**Confidence:** 4

**Review:**

Quality:
The motivation and experimentation is sound.

Originality:
This work is a natural follow up on previous work that used group lasso for CNNs, namely learning sparse RNNs with group-lasso. Not very original, but nevertheless important.

Clarity:
The fact that the method is using a group-lasso regularization is hidden in the intro section and only fully mentioned in section 3.2 I would mention that clearly in the abstract.

Significance:
Leaning small models is important and previous sparse RNN work (Narang, 2017) did not do it in a structured way, which may lead to slower inference step time. So this is an investigation of interest for the community.

Minor comments:
- One main claim in the paper is that group lasso is better than removing individual weights, yet not experimental evidence is provided for that.
- The authors found that their method beats "direct design". This is somewhat unintuitive, yet no explanation is provided.

---

> ### Author Response · Authors · 2017-12-20
> **Response to reviews**
>
> Thanks for reviewing.
>
> To clarity:
> We have mentioned group Lasso in the abstract. However, please note that, any structured sparsity optimization can be integrated into ISS, like group connection pruning based on the norm of the group (as used by Hao Li et. al. 2017 ).
>
> To minor comments:
> - The speedup vs sparsity is added in Fig. 1, to quantitatively justify the gain of structured sparsity over non-structured sparsity.
> - In our context, "direct design" refers to using the same network architecture but with smaller hidden sizes. The comparison is in Table 1.
> - We are working on a better ptb baseline -- the RHN model (https://arxiv.org/abs/1607.03474), to solve the concerns on ptb dataset. The training takes time, but we will post our results as soon as the experiments are done. However, our results on SQuAD may reflect that the approach works in general.
>
> Thanks!

---

> > ### Author Response · Authors · 2017-12-24
> > **A better ptb baseline -- the RHN model -- is added**
> >
> > To follow up the concerns on the ptb dataset.
> >
> > We have added state-of-the-art model for ptb to show the generalizability of our approach. We select Recurrent Highway Networks. Please refer to Table 1 in the paper (https://arxiv.org/pdf/1607.03474.pdf) for the state-of-the-art models on ptb. "Variational RHN + WT" is the one we used as the baseline.
> > Our results are covered in Table 2 in our paper. In a nutshell, our approach can reduce the RHN width (the number of units per layer) from 830 to 517 without losing perplexity.

---

> > > ### Comment · AnonReviewer2 · 2018-01-12
> > > **Thanks for clarifications and RHN baseline.**
> > >
> > > Increasing my score as this strengthens the work.

---

### Official Review · AnonReviewer1 · 2017-11-26
**the text is verbose but method is simple**

**Rating:** 6
**Confidence:** 4

**Review:**

The paper spends lots of (repeated)  texts on motivating and explaining ISS. But the algorithm is simple, using group lasso to find components that are can retained to preserve the performance.  Thus the novelty is limited.

The experiments results are good.

Sec 3.1 should be made more concise.

---

> ### Author Response · Authors · 2017-12-20
> **Response to reviews**
>
> Thanks for reviewing.
>
> We have made sec 3.1 as concise as possible. We have moved some to the Appendix A.
>
> The key novelty/contribution of the paper is to identify the structure inside RNNs (including LSTMs and RHNs) that shall be considered as a group ("ISS") to most effectively explore sparsity.  Once the group is identified, using group lasso becomes intuitive.  That is why we describe ISS, the structure of the group, in details and illustrate the intuitions and analysis behind it.  We clarified this in the revision.

---

### Official Review · AnonReviewer3 · 2017-11-27

**Rating:** 7
**Confidence:** 4

**Review:**

The authors propose a technique to compress LSTMs in RNNs by using a group Lasso regularizer which results in structured sparsity, by eliminating individual hidden layer inputs at a particular layer. The authors present experiments on unidirectional and bidirectional LSTM models which demonstrate the effectiveness of this method. The proposed techniques are evaluated on two models: a fairly large LSTM with ~66.0M parameters, as well as a more compact LSTM with ~2.7M parameters, which can be sped up significantly through compression.
Overall this is a clearly written paper that is easy to follow, with experiments that are well motivated. To the best of my knowledge most previous papers in the area of RNN compression focus on pruning or compression of the node outputs/connections, but do not focus as much on reducing the computation/parameters within an RNN cell. I only have a few minor comments/suggestions which are listed below:

1. It is interesting that the model structure where the number of parameters is reduced to the number of ISSs chosen from the proposed procedure does not attain the same performance as when training with a larger number of nodes, with the group lasso regularizer. It would be interesting to conduct experiments for a range of \lambda values: i.e., to allow for different degrees of compression, and then examine whether the model trained from scratch with the “optimal” structure achieves performance closer to the ISS-based strategy, for example, for smaller amounts of compression, this might be the case?

2. In the experiment, the authors use a weaker dropout when training with ISS. Could the authors also report performance for the baseline model if trained with the same dropout (but without the group LASSO regularizer)?

3. The colors in the figures: especially the blue vs. green contrast is really hard to see. It might be nicer to use lighter colors, which are more distinct.

4. The authors mention that the thresholding operation to zero-out weights based on the hyperparameter \tau is applied “after each iteration”. What is an iteration in this context? An epoch, a few mini-batch updates, per mini-batch? Could the authors please clarify.

5. Clarification about the hyperparameter \tau used for sparsification: Is \tau determined purely based on the converged weight values in the model when trained without the group LASSO constraint? It would be interesting to plot a histogram of weight values in the baseline model, and perhaps also after the group LASSO regularized training.

6. Is the same value of \lambda used for all groups in the model? It would be interesting to consider the effect of using stronger sparsification in the earlier layers, for example.

7. Section 4.2: Please explain what the exact match (EM) and F1 metrics used to measure performance of the BIDAF model are, in the text.

Minor Typographical/Grammatical errors:
- Sec 1: “... in LSTMs meanwhile maintains the dimension consistency.” → “... in LSTMs while maintaining the dimension consistency.”
- Sec 1: “... is public available” → “is publically available”
- Sec 2: Please rephrase: “After learning those structures, compact LSTM units remain original structural schematic but have the sizes reduced.”
- Sec 4.1: “The exactly same training scheme of the baseline ...” → “The same training scheme as the baseline ...”

---

> ### Author Response · Authors · 2017-12-20
> **Response to reviews**
>
> Thanks for reviewing.
>
> 1. ISS approach can learn an “optimal” structure whose accuracy is better than the same “optimal” model but trained without using group Lasso.
> For example, in Table 1, the first model learned by ISS approach has “optimal” structure with hidden sizes of (373, 315), and its perplexity is better than the same model (in the last row) but trained without using group Lasso regularization.
>
> 2. With the same dropout (keep ratio 0.6), the baseline model overfits, and, with early stop, the best validation perplexity is 97.73 which is worse than the original 82.57.
>
> 3. Changed green to yellow.
>
> 4. Per mini-batch
>
> 5. Yes, \tau is determined purely based on the trained model without group LASSO regularization. No training is needed to select it.
> Thanks for sharing this thought. Histogram is added in Appendix C. Instead of plotting the histogram of all weights, we plot the histogram of vector lengths of each “ISS weight groups”. We suppose it is more interesting because group Lasso essentially squeezes the length of each vector. The plot shows that the histogram is shifted to zeros by group Lasso regularization.
>
> 6. Yes, to reduce the number of hyper-parameters, an identical \lambda is used for all groups.
> We tried to linearly scale the strength of regularization on each group by the vector length of the “ISS group weight” as used by Alvarez et al. 2016, however, it didn’t help to improve sparsity in our experiments.
>
> 7. We now add the reference of the definition of EM and F1 (Rajpurkar et al. 2016) into the paper:
> “Exact match. This metric measures the percentage of predictions that match any one of the ground truth answers exactly.”
> “(Macro-averaged) F1 score. This metric measures the average overlap between the prediction and ground truth answer. We treat the prediction and ground truth as bags of tokens, and compute their F1. We take the maximum F1 over all of the ground truth answers for a given question, and then average over all of the questions.”
>
> 8. Corrected. Thanks for so many useful details.
>
> Paper are revised based on the comments.

---

### Public Comment · ~Aaron_Jaech1 · 2017-10-25
**choice of baseline for language modeling experiments**

If you are going to use the PTB dataset for your language modeling experiments, it would help if you use a newer baseline than the 2014 Zaremba paper. It would be better to cite "On the State of the Art of Evaluation in Neural Language Models" from July 2017. (https://arxiv.org/pdf/1707.05589.pdf) They report a perplexity of 59.6 using a single-layer LSTM with 10 million parameters.

---

> ### Author Response · Authors · 2017-12-24
> **State-of-the-art experiments on ptb added**
>
> Hi Aaron,
>
> We have added state-of-the-art model for ptb to show the generalizability of our approach. In limited time, we select Recurrent Highway Networks for fast evaluation since it is open source here https://github.com/julian121266/RecurrentHighwayNetworks. You may refer to Table 1 in the paper (https://arxiv.org/pdf/1607.03474.pdf) for the state-of-the-art models on ptb. "Variational RHN + WT" is the one we used as the baseline.
> Our results are covered in Table 2 in our paper. In a nutshell, our approach can reduce the RHN width from 830 to 517 without losing perplexity.
>
> Thanks.

---

### Decision · Program_Chairs · 2018-01-29
**ICLR 2018 Conference Acceptance Decision**

**Decision:**

Accept (Poster)

**Comment:**

The reviewers really liked this paper. This paper presents a tweak to the LSTM cell that introduces sparsity, thus reducing the number of parameters in the model.

The authors show that their sparse models match the performance of the non-sparse baselines. While the results are not state-of-the-art but vanilla implementations of standard models, this is still of interest to the community.

---

> ### Author Response · Authors · 2018-01-30
> **Updating to the state-of-the-art**
>
> Thanks for accepting our paper!!! We've taken comments from the reviewers and advanced ptb models to the state-of-the-art during the rebuttal. We will manage to advance the results on SQuAD and more. Thanks for reviewing.